# Evidence for the effectiveness of minimum pricing of alcohol: a systematic review and assessment using the Bradford Hill criteria for causality

Sadie Boniface,[1] Jack W Scannell,[2] Sally Marlow[1]

▶ Prepublication history and additional material is available. To view please visit the journal (http://dx.doi.org/10.1136/bmjopen-2016-013497).

[1]Department of Psychology and Neuroscience, National Addiction Centre, Institute of Psychiatry, King's College London, London, UK
[2]School of Social and Political Science, University of Edinburgh, Edinburgh, UK

**Correspondence to**
Dr Sadie Boniface;
sadie.boniface@kcl.ac.uk

## ABSTRACT

**Objectives:** To assess the evidence for price-based alcohol policy interventions to determine whether minimum unit pricing (MUP) is likely to be effective.

**Design:** Systematic review and assessment of studies according to Preferred Reporting Items for Systematic Reviews and Meta-Analyses (PRISMA) guidelines, against the Bradford Hill criteria for causality. Three electronic databases were searched from inception to February 2017. Additional articles were found through hand searching and grey literature searches.

**Criteria for selecting studies:** We included any study design that reported on the effect of price-based interventions on alcohol consumption or alcohol-related morbidity, mortality and wider harms. Studies reporting on the effects of taxation or affordability and studies that only investigated price elasticity of demand were beyond the scope of this review. Studies with any conflict of interest were excluded. All studies were appraised for methodological quality.

**Results:** Of 517 studies assessed, 33 studies were included: 26 peer-reviewed research studies and seven from the grey literature. All nine of the Bradford Hill criteria were met, although different types of study satisfied different criteria. For example, modelling studies complied with the consistency and specificity criteria, time series analyses demonstrated the temporality and experiment criteria, and the analogy criterion was fulfilled by comparing the findings with the wider literature on taxation and affordability.

**Conclusions:** Overall, the Bradford Hill criteria for causality were satisfied. There was very little evidence that minimum alcohol prices are not associated with consumption or subsequent harms. However the overall quality of the evidence was variable, a large proportion of the evidence base has been produced by a small number of research teams, and the quantitative uncertainty in many estimates or forecasts is often poorly communicated outside the academic literature. Nonetheless, price-based alcohol policy interventions such as MUP are likely to reduce alcohol consumption, alcohol-related morbidity and mortality.

## Strengths and limitations of this study

- This review adds to an emerging literature of systematic reviews synthesising findings using the Bradford Hill criteria for causality in research areas where traditional meta-analyses of randomised controlled trials are not possible or appropriate.
- A range of study designs were included, allowing for a comprehensive review of a disparate evidence base to investigate whether minimum unit pricing of alcohol is likely to reduce alcohol consumption and alcohol-related harm.
- Studies examining the effects of alcohol taxation or changes in alcohol affordability, or studies solely reporting on price elasticity of demand, were not included.
- Methodological quality of studies was variable.

## INTRODUCTION

There are many policies and programmes that aim to reduce harms from alcohol.[1] One of these is minimum alcohol pricing, which exists in a number of countries around the world. The most notable example of this is Canada, where there are government monopolies on alcohol sales and a variety of types of minimum pricing in operation. For example, there is a minimum price per litre of a particular drink in British Columbia[2] and a (higher) minimum price linked to drink type and strength in Saskatchewan.[3] Other countries with minimum alcohol pricing include Belarus, Kyrgyzstan, the Republic of Moldova, the Russian Federation and Ukraine.[4] Minimum alcohol pricing is being considered by governments in Ireland[5] and has also been reviewed in Australia[6] and New Zealand.[7]

The situation with regards to minimum alcohol pricing in the UK is complex.

In England and Wales, there has been a ban on alcohol being sold at below cost (the total amount of 'duty plus value added tax (VAT)') since May 2014;[8] and the first conviction for selling alcohol below this level took place in 2016.[9] Duty plus VAT is equivalent to a 70 cl bottle of vodka (37.5% alcohol by volume (ABV)) costing a minimum of £8.72,[10] whereas under a minimum price of 50 pence per unit (one UK unit=10 mL or 8 g ethanol), this would cost £13.13. In 2012, the UK coalition government cited support for minimum unit pricing (MUP) in its alcohol strategy,[11] and legislation to have a minimum price of £0.50 per unit was passed in Scotland the same year.[4] Following the change to a Conservative majority government in 2015, it is unclear whether there is still central government support for MUP. In Scotland, the Scotch Whisky Association challenged the 2012 legislation in the Scottish Court of Session, which referred the case to the Court of Justice of the European Union (CJEU) in 2014.[12] In late 2015, the CJEU referred the case back to the Scottish courts to investigate proportionality (that the same objective cannot be met through increased taxation),[13] which could have implications for other EU countries considering MUP. In late 2016, the Scottish Court of Session ruled that MUP does not contravene EU law;[14] however, the Scotch Whisky Association then appealed to the UK Supreme Court.[15]

In light of this ongoing consideration of MUP in the UK, in this paper we assess the effectiveness of minimum alcohol price interventions to reduce alcohol-related harm. Alcohol-related harm costs the National Health Service in England £3.5 billion each year and the estimated cost to society is £21 billion per year.[16] The latest annual figures for England (population of 54 million) show over 1 million alcohol-related hospital admissions (2013/2014) and 6500 alcohol-related deaths (2013); and these figures represent increases compared with a decade previously of 115% and 10%, respectively.[16]

We systematically review the literature on the effect of price interventions or policies such as MUP on alcohol consumption, alcohol-related morbidity and mortality, and wider harms. We use the nine Bradford Hill criteria for causality as a framework with the aim of assessing the likely effectiveness of MUP as a policy to reduce alcohol consumption and alcohol-related harm.

## METHODS

A systematic literature search was performed according to Preferred Reporting Items for Systematic Reviews and Meta-Analyses (PRISMA) guidance (see figure 1 for PRISMA flow diagram and online supplementary file for excluded studies).

**Figure 1** PRISMA 2009 flow diagram of studies in this systematic review. PRISMA, Preferred Reporting Items for Systematic Reviews and Meta-Analyses.

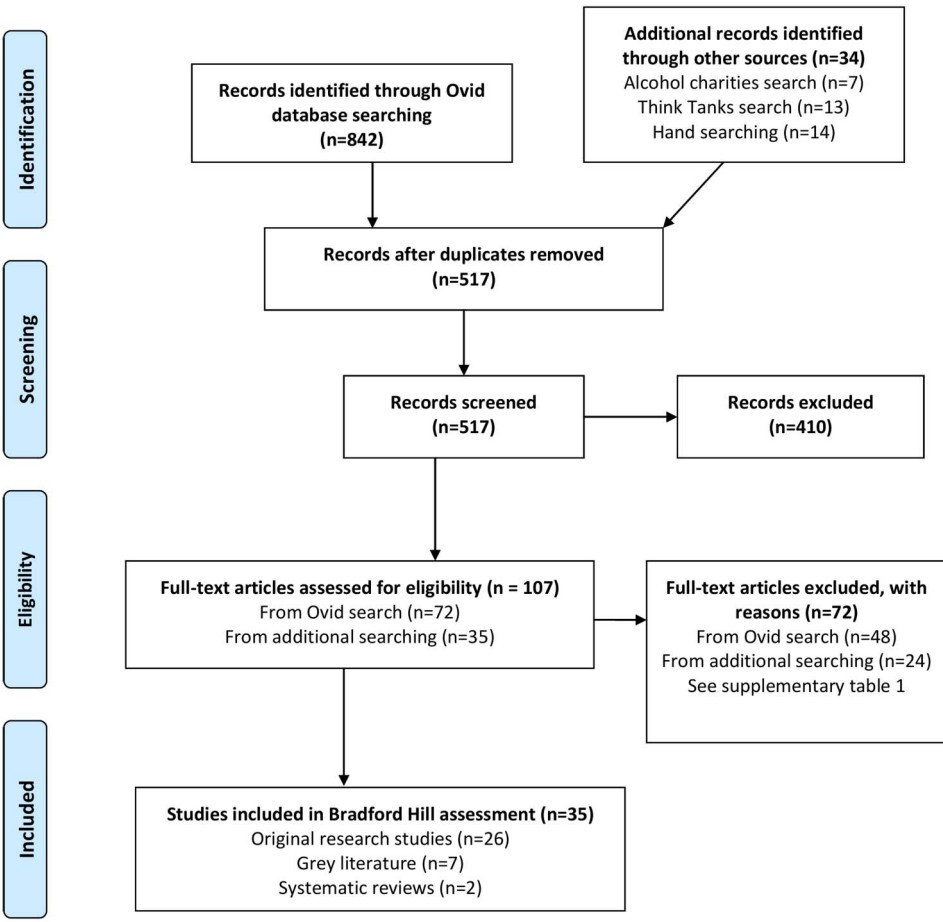

## Identification of studies

Three electronic databases were searched for titles or abstracts containing 'minimum unit pric$' OR 'minimum pric$' OR 'floor pric$' OR 'pric$ AND policy' AND alcohol. The databases were PsycINFO (1806 to February Week 2 2017), Embase (1974 to 2017 Week 07) and Ovid Medline (1946 to February Week 2 2017). We also searched the websites of five alcohol charities for publications or reports related to 'price', and also searched 20 leading UK think tanks for 'alcohol' or 'addiction'.

Inclusion criteria were: any study design; population-level studies exploring at least one aspect of the effect of interventions or policies leading to changes in the minimum price of alcohol, including but not limited to changes in alcohol sales, consumption, morbidity and mortality; and individual-level studies exploring minimum alcohol prices and alcohol purchasing, consumption, morbidity or mortality; written in English.

Exclusion criteria were: studies about taxation, affordability, price elasticity of demand for alcohol and general changes in alcohol price not the result of an intervention or policy (there is a large literature on each of these already and reviewing all of these studies was beyond the scope of this review); studies about public perceptions of MUP; and studies where a conflict of interest was reported in the paper, whether this was in favour of or against MUP.

All 33 studies that met the inclusion criteria were assessed against the Bradford Hill criteria for causality and the methodological quality appraised. These included 26 original research studies and seven studies from the grey literature, and in addition two systematic reviews pertinent to the analogy criterion were included. Of the 26 research studies, there were 9 cross-sectional surveys, 8 time series analyses or similar, 7 modelling studies, 1 qualitative study and 1 trial.

## Analysis of included studies

Quality of included studies was assessed independently by two reviewers and using validated tools. Owing to the wide variation in study designs among the included studies, the Effective Public Health Practice Project's (EPHPP) tool was used for assessing all quantitative studies, as recommended by the Cochrane Handbook for assessing studies in public health.[17] Qualitative studies (n=1) and systematic reviews (n=2) included in this review were not covered by the EPHPP tool and so were assessed using the Critical Appraisal Skills Programme (CASP) tools specific to these study designs.

Nine criteria to determine causality were suggested by Bradford Hill in an influential 1965 paper.[18] Increasingly, the Bradford Hill criteria are a standard framework to assess the impact of interventions where it is not ethical or practical to conduct randomised controlled trials (RCTs). Our interpretation of the Bradford Hill criteria for the purpose of this review is listed in table 1. Two reviewers assessed each study against each of the nine

criteria and agreed which studies provided relevant evidence for or against each criterion.

## RESULTS

The included studies that are published in peer-reviewed journals (26 research studies and two systematic reviews) are listed by study type in table 2 with information on study characteristics and methodological quality. Of the research studies, the methodological quality was rated as 'strong' in 15 studies, 'moderate' in 8 studies and 'weak' in 3 studies. Both of the systematic reviews were rated 'strong'. The seven reports from the grey literature are listed in table 3. Five of the seven were rated as of 'strong' methodological quality, with the remaining two not appropriate to rate using our critical appraisal tools.

## Consideration against the Bradford Hill criteria for determining causality
### Strength of the association

In 16/26 studies published in peer-reviewed journals, strength of the association between pricing and alcohol consumption or alcohol-related harm was evidenced by a summary statistic such as an OR, and by a test of the statistical significance of the association. As well as the statistical significance of the summary statistic, the magnitude of the effect was also considered, as a larger effect size corresponds to a greater population health impact. Studies in Canada found that 10% increases in minimum prices were associated with reductions in alcohol consumption of 3.4–8.4%,[2] [3] reductions in alcohol-attributable hospital admissions of 9%[21] and reductions in alcohol-related mortality of 32%, each of which was statistically significant.[24] Cross-sectional studies in the UK, Ireland, Australia[32–36] [38–40] and one trial from the USA[41] found statistically significant associations between cheaper alcohol and heavier drinking. The magnitude of the association varied across these studies, but due to different study measures and outcomes, the results are not all comparable. As an indication, the OR for buying alcohol below a proposed minimum price among heavier drinkers was 1.34 in Crawford's study,[35] 1.50 in Cousins's study[34] and 1.70 in Callinan's study.[33] There was not any evidence to support this criterion from the grey literature. Overall, there is reasonably good support for the strength of the association criterion.

### Consistency

This criterion requires looking across all the studies included in the review to see whether similar conclusions have been drawn. Inverse associations between alcohol pricing and alcohol consumption or harm have been documented in countries in Europe, North America and Australia, and although most studies are from the last 10 years, there are studies from the 1970s and 1980s as well. There is evidence from different research teams and different types of study including

**Table 1** Bradford Hill criteria for assessing causation and the definitions used in this review

| Criterion | Bradford Hill criteria (1965) | Application in this review |
|---|---|---|
| 1. Strength of the association | The strength of a supposed association between an intervention and an outcome is determined by the appropriate statistic used to measure the protective effect of an intervention (eg, relative risk or OR). This is the most important factor determining causation | A statistically significant change ($p < 0.05$) in alcohol consumption or alcohol-related harms, in the expected direction. The exact magnitude of the association was assessed on a study by study basis |
| 2. Consistency | Has it been repeatedly observed by different persons, in different places, circumstances and times? | Whether different studies conducted in different locations, in different populations, by different investigators and at different times have reported similar findings |
| 3. Specificity | Specificity is present when the intervention is exclusive to the outcome and when the outcome has no other known cause or associated risk factors; cautions that this criterion should not be overemphasised and that if specificity is not apparent, this does not preclude causation | If pricing was the only reason that alcohol consumption or alcohol-related harm could have fallen, this adds to the argument for causality. However, if a price intervention was one of a number of alcohol policy interventions, then this criterion is not satisfied |
| 4. Temporality | Refers to temporal relationship of association between exposure and disease outcome; to infer causality, exposure must precede outcome | The pricing intervention studied must have taken place before a change in alcohol consumption or harm was observed |
| 5. Dose–response | If the association is one in which a dose–response curve or biological gradient can be observed, this adds to the case for causality | If interventions leading to a larger increase in prices had a greater effect on alcohol consumption and alcohol-related harm than interventions where the price change was small, or if studies demonstrate that different minimum prices have differing effects, in the expected direction |
| 6. Plausibility | A likely biological mechanism linking the intervention to the observed findings helps to explain causality; plausibility depends on biological knowledge of the day | Studies that found an association between price and population-level alcohol consumption and that heavier drinkers tend to purchase the cheapest alcohol could demonstrate plausibility |
| 7. Coherence | When the evidence from different disciplines sources 'hangs well together' and does not conflict with other generally known facts, this criterion is met | Describes whether studies conducted in different settings or disciplines had complementary findings. Will not be demonstrated by a single study in isolation but rather the evidence base as a whole |
| 8. Experiment | Experimental evidence from laboratory studies or RCTs could potentially provide strongest support for causation<br>This criterion often provides the strongest support for causation and describes whether there is empirical evidence for the association | In addition to laboratory studies and RCTs, natural experiments with before-and-after measures could also show the effectiveness of minimum unit pricing in a 'real-world' setting |
| 9. Analogy | Causality is supported by analogy if there are similar associations or causal relationships in other areas of relevance, weakest form of evidence of causality | Other areas of relevance include whether higher taxation on alcohol is associated with reduced alcohol consumption and alcohol-related harm, and may require drawing on additional literature outside of the main systematic review |

RCTs, randomised controlled trials.

cross-sectional studies, time series analyses and econometric modelling studies. Support for the consistency criterion is very strong.

## Specificity

The specificity criterion relates to whether changes in alcohol consumption or harm could be attributed to anything other than the price intervention. Many studies included have statistically adjusted for confounding factors; however, the best support for the specificity criterion comes from the econometric modelling studies because there is no risk of residual confounding. The

Sheffield Alcohol Policy Model is one such model and has been applied in England,[25–29 47] Scotland[45 51] and Canada[48] and provides very strong support for the specificity criterion. Further support is provided by other different modelling studies in the UK[49 50] and Australia[30 31] and a (non-randomised) trial in the USA.[41] Thus, support for the specificity criteria is very strong.

## Temporality

It is important that pricing interventions take place before changes to alcohol consumption and harm to attribute causality. Strong support for this criterion

**Table 2**  Studies published in peer-reviewed journals included in Bradford Hill criteria assessment

| Study type | First author and year published | Country | Study design | Population or participants | Pricing intervention studied | Outcomes studied | Peer reviewed | Conflict of interest | Quality rating | Bradford Hill criteria met |
|---|---|---|---|---|---|---|---|---|---|---|
| Natural experiments and time series analyses | Bhattacharya 2013[19] | Russia | Time series analysis of panel data set | Populations of 77 Russian oblasts (provinces), 1970–2000 | Substantial increases in administratively set alcohol prices 1985–1988, along with six other antialcohol measures | Mortality | Yes | Not stated | Strong | SA, CON, TE, PL, CO, EX, |
| | Herttua 2015[20] | Finland | Time series analysis | General population using population registry | Modelled 1% increase in the average minimum price of all alcoholic beverages based on actual price increases adjusted for inflation using Consumer Price Index | Alcohol-related mortality | Yes | None | Strong | SA (not universal findings—subgroup only), CON (counter findings) TE, PL, CO, EX |
| | Stockwell 2012[2] | Canada | Cross-sectional versus time series analysis of ecological data | Population of British Columbia | Actual minimum price increased over a 20-year period. Study modelled a 10% increase in the average minimum price of all alcoholic beverages adjusted by monthly Consumer Price Index | Alcohol consumption (measured by sales) | Yes | None | Strong | SA, CON, TE, DR, CO, EX |
| | Stockwell 2012[3] | Canada | Cross-sectional versus time series analysis of ecological data | Population of Saskatchewan | Actual minimum price increased over a 7-year period. Study modelled a 10% increase in the average minimum price of all alcoholic beverages adjusted by monthly Consumer Price Index | Alcohol consumption (measured by sales) | Yes | Not stated | Strong | SA, CON, TE, DR, CO, EX |
| | Stockwell 2013[21] | Canada | Cross-sectional versus time series analysis of ecological data | Populations of 89 geographic areas in British Columbia | Actual minimum price increased over a 20-year period. Study modelled 10% increase in the average minimum price of all alcoholic beverages adjusted by monthly Consumer Price Index | Alcohol-attributable hospital admissions | Yes | Not stated | Strong | SA, CON, TE, DR, PL, CO, EX |
| | Treisman 2010[22] | Russia | Secondary analysis of historical data with focus on price changes 1990–1994 | Population of Russia | Price liberalisation of vodka in early 1990s—in 1993, real price of vodka was around 25% of that in 1990 | Mortality | Yes | Not stated | Strong | SA, CON, TE, PL, CO, EX |

Continued

**Table 2** Continued

| Study type | First author and year published | Country | Study design | Population or participants | Pricing intervention studied | Outcomes studied | Peer reviewed | Conflict of interest | Quality rating | Bradford Hill criteria met |
|---|---|---|---|---|---|---|---|---|---|---|
| | | | | | | | **Study assessment** | | | |
| | Wald 1984[23] | Poland | Analysis of routine data 1970–1981 | Population of Poland | Poor harvest led to high prices, rationing and illegal sales | Alcohol consumption and alcohol-related hospital admissions | Yes | Not stated | Weak | CON, TE, PL, CO, EX |
| | Zhao 2013[24] | Canada | Cross-sectional versus time series analysis of ecological data | Populations of 16 health service delivery areas in British Columbia, Canada | Actual minimum price increased over a 20-year period. Study modelled 10% increase in the average minimum price of all alcoholic beverages adjusted by monthly Consumer Price Index. Also looked at outlet density | Acute, chronic and wholly alcohol-attributable mortality | Yes | None | Strong | SA, CON, TE, DR, PL, CO, EX |
| Modelling studies | Brennan 2014[25] | England | Modelling study using SAPM | The UK national surveys of general population (subgroups of moderate, harmful, hazardous) | MUP of £0.40, £0.45 and £0.50. Ban on below cost selling | Alcohol consumption, consumer spending, 47 health harms, QALYs | Yes | None | Strong | CON, SP, DR, PL, CO |
| | Holmes 2014[26] | England | Modelling study using SAPM | The UK national surveys of general population (subgroups of moderate, harmful, hazardous) | MUP of 45p | Alcohol consumption, consumer spending, 47 health harms, QALYs | Yes | None | Strong | CON, SP, PL, CO |
| | Meier 2009[27] | The UK | Modelling study using SAPM | The UK national surveys of general population (subgroups of moderate, harmful, hazardous) | Ten pricing policy options, including different levels of MUP (of 33 analysed) | Alcohol consumption, consumer spending, 47 health harms, crime, employment | Yes | None | Strong | CON, SP, DR, PL, CO |
| | Meier 2016[28] | England | Modelling study using SAPM | The UK national surveys of general population (subgroups of moderate, increasing risk, heavy) | MUP of £0.50 compared with three alcohol taxation interventions | Alcohol consumption in different income and socioeconomic groups | Yes | None | Strong | CON, SP, PL, CO |
| | Purshouse 2010[29] | England | Modelling study using SAPM | The UK national surveys of general population (subgroups of moderate, harmful, hazardous) | 18 different pricing policies (including MUP) | Alcohol consumption, consumer spending, 47 health harms, QALYs | Yes | None | Strong | CON, SP, DR, PL, CO |

Continued

**Table 2** Continued

| Study type | First author and year published | Country | Study design | Population or participants | Pricing intervention studied | Outcomes studied | Peer reviewed | Conflict of interest | Quality rating | Bradford Hill criteria met |
|---|---|---|---|---|---|---|---|---|---|---|
| | | | | | | | | | | |
| | Sharma 2016[30] | Australia | Counterfactual analysis | Representative sample of households (n=884) completing 12-month Homescan shopping survey | MUP of A$2 | Alcohol purchasing and consumption | Yes | None | Strong | CON, SP, PL, CO |
| | Vandenberg 2016[31] | Australia | Modelling study | Representative sample of households (n=885) completing Homescan shopping survey | MUP of A$1 compared with a specific alcohol tax | Alcohol purchasing and consumption | Yes | None | Strong | CON, SP, PL, CO |
| Cross-sectional studies | Black 2011[32] | Scotland | Cross-sectional survey | 377 hospital patients with serious alcohol problems | The UK alcohol units purchased below proposed MUP of £0.40p/£0.50p | Alcohol consumption | Yes | None | Moderate | SA CON, DR, PL, CO |
| | Callinan 2015[33] | Australia | Cross-sectional survey | Drinkers 18+ participating in Australian International Alcohol Control study (n=1681) | Australian standard drinks purchased below proposed minimum prices of A$0.80/A$1.00/A$1.25 | Alcohol consumption | Yes | Not stated | Moderate | SA, CON, DR, PL, CO |
| | Cousins 2016[34] | Ireland | Cross-sectional survey | 3187 adults in 2013 National Alcohol Diary Survey | Alcohol units purchased below proposed minimum price of €1.00 | AUDIT-C score | Yes | None | Strong | SA, CON, PL, CO |
| | Crawford 2012[35] | England | Cross-sectional survey | 515 members of the public | The UK alcohol units purchased below proposed MUP of £0.50 | AUDIT score | Yes | None | Moderate | SA, CON, PL, CO |
| | Falkner 2015[36] | New Zealand | Cross-sectional survey | 115 adults undergoing alcohol detoxification | New Zealand standard drinks purchased below proposed minimum prices of NZ$1.00/NZ$1.10/NZ$1.20 | Alcohol consumption | Yes | No | Moderate | SA, CON, PL, CO |
| | Forsyth 2014[37] | Scotland | Cross-sectional survey | Shopkeepers of 144 off licences in Glasgow | MUP of £0.50 | Products affected and hospital admissions | Yes | None | Weak | CON, PL (weakly), CO |
| | Ludbrook 2012[38] | The UK | Cross-sectional survey | Expenditure and Food Survey data from 20062008 (n=18 624) | Purchasers of alcohol < £0.45 per unit | Income of purchasers of cheap alcohol | Yes | Not stated | Moderate | SA, CON, PL, CO |
| | Sharma 2014[39] | Australia | Cross-sectional survey | Representative sample of households (n=885) completing shopping survey | MUP of A$1 and taxation | Alcohol consumption (measured by projected sales) | Yes | None | Moderate | SA, CON, DR, PL, CO |

Continued

**Table 2** Continued

| Study type | First author and year published | Country | Study design | Population or participants | Pricing intervention studied | Outcomes studied | Peer reviewed | Conflict of interest | Quality rating | Bradford Hill criteria met |
|---|---|---|---|---|---|---|---|---|---|---|
| | | | | | | | | | | |
| | Sheron 2014[40] | The UK | Cross-sectional survey | Adult patients in a liver unit of a hospital (n=204) | The UK alcohol units purchased below £0.50 | Alcohol consumption | Yes | Not stated | Moderate | SA, CON, DR, PL, CO |
| Intervention studies | Babor 1978[41] | The USA | Trial (not randomised) | 34 male volunteers in live-in research facility | 'Happy hour' with a reduction in set price of alcohol for one group of participants | Alcohol consumption | Yes | Not stated | Weak | SA, CON, SP, TE, CO, EX |
| Qualitative studies | Seaman 2013[42] | Scotland | Qualitative study | 130 participants aged 16–30 | Hypothetical minimum price increases | Alcohol consumption and substitution with other substances | Yes | None | Moderate | CON, CO |
| Systematic reviews | Wagenaar 2009[43] | Worldwide | Systematic review and meta-analysis | Studies tended to cover general population | Alcohol price and taxation interventions studied together | Alcohol consumption (measured by alcohol sales or self-reported consumption) | Yes | None | Strong | AN |
| | Wagenaar 2010[44] | Worldwide | Systematic review and meta-analysis | Studies tended to cover general population | Alcohol price and taxation interventions studied together | Alcohol-related morbidity (disease, injury, suicide, traffic crashes, sexually transmitted diseases, other drug use, crime and misbehaviour) and mortality | Yes | Not stated | Strong | AN |

Abbreviations for the Bradford Hill criteria: AN, analogy; CO, coherence; AUDIT, Alcohol Use Disorders Identification Test; CON, consistency; DR, dose–response; EX, experiment; MUP, Minimum Unit Pricing; PL, plausibility; SA, strength of the association; SAPM, Sheffield Alcohol Policy Model; SP, specificity; TE, temporality; QALYs, Quality Adjusted Life Years.

**Table 3** Studies published in the grey literature included in Bradford Hill criteria assessment

| Author and year published | Study characteristics | | | | | Study assessment | | | Bradford Hill criteria met |
|---|---|---|---|---|---|---|---|---|---|
| | Country | Study design | Population or participants | Pricing intervention studied | Outcomes studied | Peer reviewed | Conflict of interest | Quality rating | |
| Angus 2016[45] | Scotland | Modelling study using SAPM | Scottish general population survey (subgroups of moderate, harmful, hazardous) | MUP of 30p, 40p, 50p, 60p and 70p, compared with taxation interventions | Alcohol consumption, consumer spending, exchequer and retail revenue, 47 health harms | Not stated | None | Strong | CON, SP, DR, PL, CO |
| Booth 2008[46] | Worldwide | Review of reviews and systematic review | Studies tended to cover general population | Various minimum unit prices and taxation interventions | Alcohol consumption and various measures of alcohol harm | Yes | None | Strong | AN |
| Brennan 2008[47] | England | Modelling study using SAPM | Adults in England | General price increases. MUP of £0.20, £0.25, £0.30, £0.35, £0.40, £0.45, £0.50, £0.60 and £0.70. Restrictions on off-trade price promotions. | Alcohol consumption, consumer spending, sales duty and VAT, 47 health harms, crime and employment | Not stated | None | Strong | CON, SP, DR, PL, CO |
| Hill McManus 2012[48] | Canada | Modelling study using SAPM | Adults in two Canadian provinces (Ontario and British Columbia) | MUP of C$1.50 | Alcohol consumption, consumer spending, hospital admissions, mortality, crime | No | None | Strong | CON, SP, PL, CO |
| Institute for Fiscal Studies 2010[49] | Great Britain | Economic modelling study using market research data | Shopping data from 25 248 British households | MUP of £0.45 | Alcohol consumption | Not stated | Not stated | Not possible to rate | CON, SP, CO |
| Institute for Fiscal Studies 2013[50] | Great Britain | Economic analysis | Population of Great Britain | MUP of £0.45 and increased alcohol taxation | Alcohol consumption | Not stated | Not stated | Not possible to rate | CON, SP, CO |
| Meng 2010[51] | Scotland | Modelling study using SAPM | Adults in Scotland | MUP of £0.20, £0.25, £0.30, £0.35, £0.40, £0.45, £0.50, £0.60 and £0.70. Restrictions on off-trade price promotions. | Alcohol consumption, consumer spending, 47 health harms, crime, employment | Not stated | None | Strong | CON, SP, DR, PL, CO |

AN, analogy; CO, coherence; CON, consistency; DR, dose–response; EX, experiment; PL, plausibility; SA, strength of the association; SAPM, Sheffield Alcohol Policy Model; SP, specificity; TE, temporality.

comes from research following the introduction of MUP in Canada, where minimum price increases preceded reductions in alcohol consumption,[2] [3] alcohol-attributable hospital admissions[21] and alcohol-related mortality.[24] Studies where price changes preceded the expected changes in alcohol consumption or harm have also been conducted in Russia,[19] [22] Poland[23] and Finland.[20] Overall, there is very strong support for the temporality criterion.

### Dose–response/biological gradient

This criterion is supported if different price levels have been found to have differing effects on consumption or harm. Many of the studies using the Sheffield Alcohol Policy Model explore the impact of a range of potential MUP options,[25] [27] [29] [45] [51] and these consistently suggest that the higher the MUP the greater the reductions in alcohol consumption or alcohol-related harms. The Canadian studies of minimum pricing lend further support for this criterion because the analysis presents the effect on consumption or harm of a modelled 1% increase in price, meaning dose response can be inferred.[2] [3] [21] [24] Dose response is supported to a lesser extent by evidence from cross-sectional studies that heavier drinkers are more likely to pay less than a proposed MUP.[32–34] [39] [40] Overall, there is strong support for the dose–response criterion, although the relationship is difficult to quantify.

### Plausibility

This criterion refers to whether there is evidence that alcohol price can be used as an economic mechanism to influence consumption at a population level, and whether heavy drinkers tend to purchase cheaper alcohol. There is evidence from 21/26 research studies and 4/7 studies in the grey literature that the price of alcohol is inversely related to alcohol-related morbidity, hospital admissions or mortality. Moreover, there is also evidence from numerous cross-sectional studies in the UK, Ireland and Australia[32–36] [38–40] and one trial from the USA[41] that heavier drinking was significantly associated with purchasing alcohol below specified prices, further suggesting that economic mechanisms such as minimum pricing would particularly affect the heaviest drinkers. This provides strong support for the plausibility criterion.

### Coherence

This criterion refers to whether studies from different disciplines have had complementary findings and whether these fit or 'hang' well together. It is different to consistency, which is more concerned with reproducibility of findings. The findings of the majority of studies supported the coherence criterion in that they suggest that real-world MUP[2] [3] [21] [24] or minimum price increases[19] [20] [23] led to reductions in alcohol consumption and alcohol-related harm and cross-sectional surveys find that it is the heavier drinkers that are

drinking the cheapest alcohol.[32] [40] The modelling studies which use survey data in turn suggest heavier drinkers will be most affected by MUP.[26] Overall, the evidence base provides strong support for this criterion.

### Experiment

We have not identified any RCTs of minimum pricing or price-based interventions to reduce alcohol consumption. There is a small (and not randomised) trial from the 1970s[41] which found participants living in controlled conditions and offered a daily 'happy hour' discount drank significantly more alcohol than those who were not offered the discount. There is, however, substantial evidence in support of the experiment criterion from time series analyses or natural experiments, for example, where minimum pricing was introduced in Canada[3] [21] [24] and where prices fluctuated in the late 1980s and early 1990s in Russia,[19] [22] and to a lesser extent in Finland, where minimum price increases were associated with reduced mortality only among men with a basic education.[20] These studies provide tentative support for the experiment criterion.

### Analogy

To address the analogy criterion, areas related to minimum alcohol pricing must be considered. There is evidence from literature on the affordability of alcohol[52] that consumption and harm are very responsive to the affordability of alcohol. Large systematic reviews have investigated the price elasticity of demand for alcohol,[53] and have found that higher alcohol pricing and taxation (considered together) are associated with reductions in alcohol consumption, alcohol-related morbidity and mortality.[43] [44] [46] Overall, the support for the analogy criterion is very strong, although Bradford Hill describes this as the weakest evidence for causality.

### DISCUSSION

We assessed 26 research studies and two systematic reviews, plus a further seven studies from the grey literature in this review of the evidence for priced-based interventions—such as MUP—to reduce alcohol consumption and alcohol-related harm. All nine of the Bradford Hill criteria for causality were met and the vast majority of studies offered support for price-based alcohol policy interventions. However, the evidence for two of the criteria, although present, was not as strong as it was for the other criteria. These criteria were strength of the association (criterion 1) and experiment (criterion 8), and according to Bradford Hill, these are the two criteria that can provide the strongest evidence for causality. Therefore, although all of the criteria were supported, we conclude that it is highly probable, but not definite, that introducing MUP for alcohol would reduce alcohol consumption and alcohol-related harms. It is also of note that different types of study tended to satisfy different Bradford Hill criteria, and that different

study designs also produced evidence of the effectiveness of minimum pricing in relation to different outcomes. This is summarised in figure 2. This underlines the importance of including a variety of study designs in this review.

Strengths of this study are that this is the first to have systematically reviewed the literature relevant specifically to alcohol minimum pricing policies. We had broad inclusion criteria with regards to study design, price intervention and outcome measure, allowing for a comprehensive review of the evidence base. Application of the Bradford Hill criteria as part of a narrative systematic literature review is a useful and emergent technique for identifying causality: a PubMed search for systematic reviews with 'Bradford Hill' mentioned in the title or abstract yielded 28 results, 90% of which were published in the last 5 years. The limitations of this systematic review relate mainly to the broad range of studies included. It was not possible to conduct any kind of meta-analysis and therefore we do not present a pooled estimate for the likely effect of MUP on certain outcomes. The exact effect of any MUP would be influenced by a range of factors, including: the minimum price level chosen, how broadly it is applied, how strongly it is enforced and contextual factors such as affordability (in the UK, alcohol was 54% more affordable in 2014 than it was in 1980[16]), other governmental regulations and the price-level pre-MUP. Occasionally, minimum pricing has been implemented as part of a range of measures,[19] and these studies were considered alongside studies where MUP was implemented in isolation. This emphasises the importance of the specificity criterion.

There were also challenges with the quality appraisal. The EPHPP quality assessment tool was used to assess quantitative studies and the majority of studies were rated as strong or moderate. However, it was not possible to appraise two of the studies from the grey literature using this tool, and there were some challenges assessing the econometric modelling studies against this framework. However, overall we think that our quality appraisal across the different studies is broadly comparable. It should also be noted that although a number of studies were rated as 'strong', this is in relation to their respective study designs and does not reflect the position of the study type in the hierarchy of evidence framework.

This is the first systematic review that has addressed the effectiveness of minimum alcohol price interventions such as MUP using the Bradford Hill criteria. It was beyond the scope of this review to study the impact of generalised increases in alcohol prices (as opposed to minimum prices). However, where such studies have been carried out, a minimum price or floor price has been recommended, for example, in Gruenewald's 2006 study in Sweden which found that the lowest quality (the cheapest) alcohol has the highest price elasticity.[54] Previous systematic reviews of alcohol price and consumption[43] and alcohol-related harm[44] have tended to consider the effect of price increases and increased taxation together. These reviews found significant effects on consumption and morbidity and mortality. Although price regulation and taxation are closely related policy options, evidence from surveys[55] and modelling studies[45] suggests that the effects of each are different, although it is known that the majority of tax increases are passed on as increased prices for consumers.[43 56] It was beyond the scope of this review to discuss whether MUP is regressive in detail, but as it only affects the prices of the cheapest drinks, which are usually consumed by the heaviest drinkers, MUP is likely to narrow health inequalities.[28 31] A recent rapid evidence review

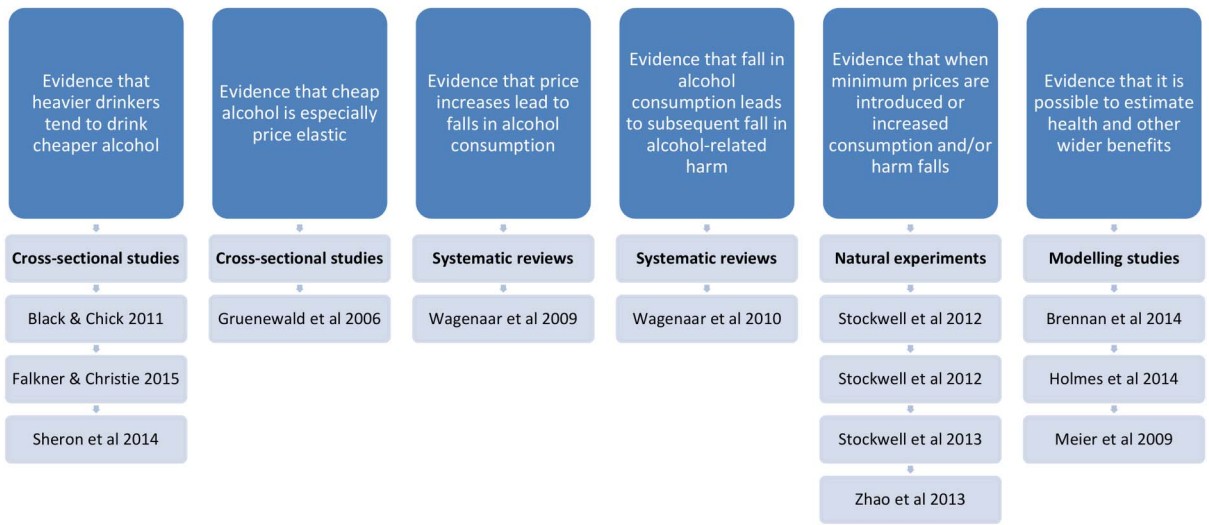

**Figure 2** This model shows that different study types tended to produce evidence of effectiveness of minimum pricing in relation to different outcomes. Studies cited in the figure are key examples of the literature in that area and do not represent an exhaustive list.

published in The Lancet examined alcohol control policies in England and recommended a combination of MUP and tax increases to reduce alcohol harm and increase government revenue, rather than either in isolation.[57] It is also important to highlight that a considerable proportion of included studies were produced by a small number of research teams. Also, with regards to the econometric modelling studies, uncertainty in estimates or forecasts is often poorly communicated outside of the academic literature. The overall risk of bias in the included studies was minimised by excluding studies with a conflict of interest (either for or against MUP). It was not possible to assess publication bias using an analytical technique such as a funnel plot due to the narrative nature of the review; however, we anticipate that by including grey literature in this review, we have mitigated publication bias as far as reasonably possible.

Overall, the findings of this review lend strong support for policies such as MUP in reducing alcohol consumption and alcohol-related harm, with all nine of the Bradford Hill criteria met, and little by way of counter findings. As it is unlikely to be feasible to conduct RCTs of MUP, the decision whether or not to introduce MUP will not be informed by a systematic review and meta-analysis of RCTs, and therefore, this synthesis of evidence according to the Bradford Hill criteria is of value.

Unanswered questions about the effectiveness of MUP remain; for example, this review has highlighted that support was moderate or tentative for two of the Bradford Hill criteria ('strength of the association' and 'experiment', respectively). There may be opportunities to explore this in countries such as Scotland if MUP is implemented. If Scotland were to implement MUP, then it would be possible to evaluate the validity of the Sheffield Alcohol Policy Model studies conducted using Scottish data. It would also be possible to conduct a longitudinal study to evaluate the effectiveness of MUP in reducing alcohol consumption and alcohol-related morbidity and mortality. The findings of this natural experiment would have relevance elsewhere within and outside the UK.

**Acknowledgements** We would like to thank Rebecca McDonald for advice on using the Bradford Hill criteria in a systematic review, Dr James Nicholls for advice on study interpretation and Dr Daniel Stahl for statistical advice on some of the included studies.

**Contributors** SM conceived the idea. SB conducted the initial search. SM and SM contributed to independently reviewing abstracts, hand-searching reference lists, completing data extraction and conducting quality appraisal. All authors contributed to the analysis and interpretation of the results and contributed to writing the manuscript. SB is guarantor.

**Funding** Open access for this article was funded by King's College London Open Scholarship Fund.

**Competing interests** SB and SM work at King's College London, which as an institution is listed as a member of the Alcohol Health Alliance. SM has received funding indirectly from UKCTAS, which as an institution is also listed as a member of the Alcohol Health Alliance. None of the authors have any relationship with the Alcohol Health Alliance.

**Provenance and peer review** Not commissioned; externally peer reviewed.

**Data sharing statement** No additional data are available.

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
