## [Reviewer comments · BMJ Open]

ARTICLE DETAILS

TITLE (PROVISIONAL)	Evidence for the effectiveness of minimum pricing of alcohol: a systematic review and assessment using the Bradford Hill criteria for causality
AUTHORS	Boniface, Sadie; Scannell, Jack; Marlow, Sally

VERSION 1 - REVIEW

REVIEWER	Tim Stockwell Centre for Addictions Research of BC, University of Victoria, BC, Canada
REVIEW RETURNED	01-Sep-2016

GENERAL COMMENTS	General comments In my opinion, this is an excellent contribution on a highly topical public-health policy issue: the effectiveness of minimum unit pricing (MUP) as a means of reducing population levels of alcohol-related harm i.e. one of the major contributors to the global burden of disease. It is topical because several EU countries are considering adopting the policy which at present is only implemented in some form in Eastern Europe, Russia and Canada. There have been EU and UK legal processes to establish the legality of the policy and the issue of evidence for effectiveness will be pivotal in future decision making. Aside from relevance and topicality, I recommend this contribution as filling an important gap in the evidence-base as it will constitute the first systematic review specifically on the evidence for minimum alcohol pricing. The research literature on this topic is fairly new with almost no studies prior to 2008. The authors have followed PRISMA guidelines, identified a surprisingly large amount of relevant literature, summarised it clearly and usefully as well as subjected it to quality assessment. As someone who has largely worked in the sphere of evaluating population level public health policies for which it is not possible to conduct randomised controlled trials, I'm delighted to see the authors addressing head-on how one assesses the strength of evidence in the absence of RCTs. The idea of applying the Bradford Hill criteria for causation in epidemiology is innovative and highly appropriate for dealing with this kind of evidence base. Specific comments I have a few suggestions for minor clarification or edits. 1. First paragraph line 15: I believe the Scottish MUP legislation was initially to introduce a 45p not 50p floor price, but worth checking the correct value.2. Page five, lines 31-33: I don't think it is appropriate to apply the Bradford Hill criteria on causation so literally as to discuss biological causal mechanisms. The criteria were developed for assessing
--

possible causal relationships between risk factors/exposure variables and diseases for which some kind of biological mechanism will almost invariably be implicated. In this case the criteria are being applied to assess evidence of effectiveness for a public health measure that uses an economic mechanism i.e. the economic principle that consumers are influenced by the price of products. Here I think the evidence for the mechanism by which the policy is thought to work would be a) evidence in general that alcohol prices influence consumption at the individual and population levels b) that the heaviest drinkers prefer the cheapest alcohol. The criteria are being applied not to assess whether alcohol itself causes health harms but whether the policy influences consumption and hence health. I think this distinction should be made explicitly and the term "biological" be removed from the table and the appropriate mechanism be identified as an economic one.

3. I'm curious about some of the studies that have been included that evaluate price changes. Some are clearly only about pricing in general (e.g. the Wagenaar systematic review) and others do not appear to be explicitly about the impact of minimum pricing (e.g. Bhattachery et al). To the extent that we need evidence on price in general, I think it would be sufficient to just refer to recent systematic reviews without doing a systematic search for these. There are many to choose from. If a study is evaluating the effect of a price change only, not of a minimum price change, I think it should be excluded from this review (e.g. Byrnes et al). There is at least one other paper that could usefully have been included that actually does (without directly naming it) evaluate the effect of increasing the price of the cheapest alcohol. [Gruenewald PJ, Ponicki WR, Holder HD, Romelsjo A. Alcohol prices, beverage quality, and the demand for alcohol: Quality substitutions and price elasticities. *Alcoholism-Clinical and Experimental Research*. 2006; 30(1): 96-105.] This is a foundational study which shows that cheap alcohol has the highest price elasticity and, further, recommends a focus on floor prices as a priority for preventing alcohol-related harm.

4. For what it's worth, I can confirm that the Hill-McManus study was not peer-reviewed!

5. Page 18 lines 12-13: the effect sizes for minimum price changes on consumption in these studies should really be listed as ranging between 3.4% and 8.4%. These were the aggregate effects on total alcohol consumption. The higher figures given in the text (10-16%) relates to individual beverage elasticities. When the price of a single beverage goes up there is usually a compensatory switch to other cheaper beverages, so measures of effects on aggregate consumption are more appropriate when assessing the effect size for potential public health impacts.

6. Page 19 lines 25-35: again I recommend removing the term "biological" from this Bradford Hill criterion (with explanation of course) and inserting text about economic mechanisms plus evidence these are especially powerful for heavier drinkers.

7. Page 19 lines 51-52: it is said that affordability was shown to be more important than price, however because affordability is the product of price and income it should be clarified whether what is actually being said is that income is more important than price. Otherwise it simply saying that price plus income is more important than price alone which is an important distinction to make as price is still very much in the mix.

8. It occurs to me that there is an underlying logic behind the relevance of each type of study that might usefully be illustrated with a logic model. It might go something like this: (i) evidence that price in general reduces alcohol consumption (ii) reduced alcohol

	consumption for individuals and populations produces alcohol-related harm (iii) evidence that heavier drinkers tend to drink cheaper alcohol (lots of cross-sectional studies) (iv) evidence that cheap alcohol is especially price elastic (v) evidence that when these assumptions are applied to existing data it is possible to estimate health and other benefits from the intervention (modelling studies) (vi) evidence that when minimum prices are increased or introduced de novo consumption and/or harm is reduced. 9. Along the same lines, I suggest that the excellent Table 2 be organised by study type or the type of evidence it contributes, perhaps in a similar order to the above e.g. cross-sectional studies relating drinker types to prices paid/modelling studies/time series studies of minimum price of effectiveness. Qualitative studies are another category. I would keep the evidence for effectiveness of pricing in general out of the table and refer to it only in the text as context - it certainly has not been dealt with systematically in the review otherwise.
--	--

REVIEWER	Norman Giesbrecht Centre for Addiction & Mental Health; Dalla Lana School of Public Health, University of Toronto
REVIEW RETURNED	22-Sep-2016

GENERAL COMMENTS	This a very timely paper, that is clearly written, and the material presented in a coherent manner. I have three general comments, and then a few specific suggestions. 1. Conflict of interest. This appears early on and stated as a rationale for excluding studies from the final analysis. What does it mean? Papers where one or more of the authors of this manuscript was a co-author on another study on alcohol pricing? Research funded by the alcohol industry or their social aspect organizations? Research funded by health departments or finance departments who would expect to gain revenue from implementing higher minimum prices on alcohol? Depending on how broader or narrow the criteria are will likely impact the number and type of studies included in the final analysis. I would prefer that those studies that meet the other criteria be included, even if there was a perceived or reported conflict of interest. 2. Dissecting minimum pricing. There are instances where the regulated minimum prices are not identical per standard drink, for beer, wine, spirits, coolers. Also, there may be minimum prices, but if pricing is not linked to pure ethanol volume then, for example, stronger beer will be cheaper per unit of pure alcohol than weaker beer. Also, in some cases minimum prices are indexed to inflation, in other cases, not. Of the 35 studies included, I would like to know how many took into account these nuances in their assessment, and how these differences were perceived to impact the outcomes. This would involve going back to the 35 studies, and adding a column to table 2, e.g., between the pricing intervention and outcomes studies column, to provide more detail, as appropriate, where the specific research provided it. 3. Other interventions. This relates to the specificity criteria. I am curious how many of the interventions assessed in the 35 papers only involved alcohol pricing, or if a number also had other policy
---

	changes underway during the period under study. I think it would be useful to highlight the latter in the Discussion, and offer some hypothesis if the other interventions enhanced or deflated the outcome. 4. Page 3, line 9. For the international audience suggest adding a phrase about the current population of England. 5. Page 3, line 23 Suggest adding a line specifically stating why the U.K. government withdrew its support for MUP.
--	---

VERSION 1 – AUTHOR RESPONSE

Reviewer 1 - Tim Stockwell

Reviewer's comment	Authors' response
General comments In my opinion, this is an excellent contribution on a highly topical public-health policy issue: the effectiveness of minimum unit pricing (MUP) as a means of reducing population levels of alcohol-related harm i.e. one of the major contributors to the global burden of disease. It is topical because several EU countries are considering adopting the policy which at present is only implemented in some form in Eastern Europe, Russia and Canada. There have been EU and UK legal processes to establish the legality of the policy and the issue of evidence for effectiveness will be pivotal in future decision making. Aside from relevance and topicality, I recommend this contribution as filling an important gap in the evidence-base as it will constitute the first systematic review specifically on the evidence for minimum alcohol pricing. The research literature on this topic is fairly new with almost no studies prior to 2008. The authors have followed PRISMA guidelines, identified a surprisingly large amount of relevant literature, summarised it clearly and usefully as well as subjected it to quality assessment. As someone who has largely worked in the sphere of evaluating population level public health policies for which it is not possible to conduct randomised controlled trials, I'm delighted to see the authors addressing head-on how one assesses the strength of evidence in the absence of RCTs. The idea of applying the Bradford Hill criteria for causation in epidemiology is innovative and highly appropriate for dealing with this kind of evidence base.	We are pleased that the reviewer sees our manuscript as relevant and topical and that they consider our methodological approach innovative and highly appropriate.
Specific comments I have a few suggestions for minor clarification or edits. 1. First paragraph line 15: I believe the Scottish MUP legislation was initially to introduce a 45p not 50p floor	We have checked the WHO report we cited, as well as the Scottish Government website and BBC news report of the legislation and believe that 50p is correct

price, but worth checking the correct value.	proposed MUP in Scotland in 2012. In the UK Government's 2012 Alcohol Strategy the proposed MUP was 40p, but this is not linked to the Scottish legislation
2. Page five, lines 31-33: I don't think it is appropriate to apply the Bradford Hill criteria on causation so literally as to discuss biological causal mechanisms. The criteria were developed for assessing possible causal relationships between risk factors/exposure variables and diseases for which some kind of biological mechanism will almost invariably be implicated. In this case the criteria are being applied to assess evidence of effectiveness for a public health measure that uses an economic mechanism i.e. the economic principle that consumers are influenced by the price of products. Here I think the evidence for the mechanism by which the policy is thought to work would be a) evidence in general that alcohol prices influence consumption at the individual and population levels b) that the heaviest drinkers prefer the cheapest alcohol. The criteria are being applied not to assess whether alcohol itself causes health harms but whether the policy influences consumption and hence health. I think this distinction should be made explicitly and the term "biological" be removed from the table and the appropriate mechanism be identified as an economic one.	We thank the reviewer for this helpful insight into applying this criterion. Although it is a very straightforward criterion to understand from a biological and risk factor perspective, we did find it challenging applying it to an economic intervention. We have removed 'biological' from the title of the criterion, rephrased our definition/application of this criterion in Table 1 and also updated our interpretation of this in the main body of the manuscript.
3. I'm curious about some of the studies that have been included that evaluate price changes. Some are clearly only about pricing in general (e.g. the Wagenaar systematic review) and others do not appear to be explicitly about the impact of minimum pricing (e.g. Bhattachery et al). To the extent that we need evidence on price in general, I think it would be sufficient to just refer to recent systematic reviews without doing a systematic search for these. There are many to choose from. If a study is evaluating the effect of a price change only, not of a minimum price change, I think it should be excluded from this review (e.g. Byrnes et al). There is at least one other paper that could usefully have been included that actually does (without directly naming it) evaluate the effect of increasing the price of the cheapest alcohol. [Gruenewald PJ, Ponicki WR, Holder HD, Romelsjo A. Alcohol prices, beverage quality, and the demand for alcohol: Quality substitutions and price elasticities. Alcoholism-Clinical and Experimental Research. 2006; 30(1): 96-105.] This is a foundational study which shows that cheap alcohol has the highest price elasticity and, further, recommends a focus on floor prices as a priority for	Gruenewald 2006 is now included in the main body of the text. We are familiar with this paper but had excluded it as we had interpreted the Swedish model as a taxation intervention (we did not include papers about tax interventions), but we have re-read the paper and now understand that this taxation works as a price intervention as this is passed on to consumers through the state alcohol monopoly. This paper is now included in the narrative but not the results of the systematic review, as suggested by the reviewer. We have re-visited the papers we have included about price changes and have now excluded five papers from the manuscript:  • Bhattacharya 2013: prices were fixed administratively (i.e. not by markets) therefore was like a minimum price –

preventing alcohol-related harm.	include  • Byrnes 2013: there was no intervention/event which caused a price change, price change indicators came from real price indices – exclude • Gilligan 2012: price did not change, was comparing relative prices in different EU countries – exclude • Sloan 1994: there was no intervention/event which caused a price change, price change indicators came from real price indices – exclude • Sutton 1995: there was no intervention/event which caused a price change, price data come from expenditure data – exclude • Treisman 2010: price liberalisation was effectively the removal of minimum/fixes prices – include • Wald 1984: restricted supply led to increased prices and rationing – include • Wall 2013: there was no intervention/event which caused a price change, price change indicators came from real price indices – exclude
4. For what it's worth, I can confirm that the Hill-McManus study was not peer-reviewed!	Thank you, we have updated Table 3 with this information
5. Page 18 lines 12-13: the effect sizes for minimum price changes on consumption in these studies should really be listed as ranging between 3.4% and 8.4%. These were the aggregate effects on total alcohol consumption. The higher figures given in the text (10-16%) relates to individual beverage elasticities. When the price of a single beverage goes up there is usually a compensatory switch to other cheaper beverages, so measures of effects on aggregate consumption are more appropriate when assessing the effect size for potential public health impacts.	We have made this change in the manuscript
6. Page 19 lines 25-35: again I recommend removing the term "biological" from this Bradford Hill criterion (with explanation of course) and inserting text about economic mechanisms plus evidence these are especially powerful for heavier drinkers.	We have made these changes in the manuscript
7. Page 19 lines 51-52: it is said that affordability was shown to be more important than price, however because affordability is the product of price and income it should be clarified whether what is actually being said is that income is more important than price. Otherwise it simply saying that price plus income is more important than price alone which is an important distinction to make as price is still very much in the	We agree with the reviewer. However we have actually now excluded this paper we were discussing regarding affordability following the recommendation above, so have removed this sentence from the manuscript.

mix.	
8. It occurs to me that there is an underlying logic behind the relevance of each type of study that might usefully be illustrated with a logic model. It might go something like this: (i) evidence that price in general reduces alcohol consumption (ii) reduced alcohol consumption for individuals and populations produces alcohol-related harm (iii) evidence that heavier drinkers tend to drink cheaper alcohol (lots of cross-sectional studies) (iv) evidence that cheap alcohol is especially price elastic (v) evidence that when these assumptions are applied to existing data it is possible to estimate health and other benefits from the intervention (modelling studies) (vi) evidence that when minimum prices are increased or introduced de novo consumption and/or harm is reduced.	We have put together a simple model to demonstrate the relevance of the different types of study, with examples, in relation to the different areas for which there is evidence. We have used the reviewers suggestions given here as a guide and welcome the reviewer's feedback on this figure (figure 2).
9. Along the same lines, I suggest that the excellent Table 2 be organised by study type or the type of evidence it contributes, perhaps in a similar order to the above e.g. cross-sectional studies relating drinker types to prices paid/modelling studies/time series studies of minimum price of effectiveness. Qualitative studies are another category. I would keep the evidence for effectiveness of pricing in general out of the table and refer to it only in the text as context - it certainly has not been dealt with systematically in the review otherwise.	We have re-arranged Table 2 to present the included studies by study design, as recommended. We have kept the effectiveness of pricing in general out of this table, as mentioned above, 5 studies have now been excluded.

Reviewer 2 – Norman Giesbrecht

Reviewer's comment	Authors' response
This a very timely paper, that is clearly written, and the material presented in a coherent manner. I have three general comments, and then a few specific suggestions.	We thank the reviewer and are pleased they consider our manuscript to be timely and clearly written
1. Conflict of interest. This appears early on and stated as a rationale for excluding studies from the final analysis. What does it mean? Papers where one or more of the authors of this manuscript was a co-author on another study on alcohol pricing? Research funded by the alcohol industry or their social aspect organizations? Research funded by health departments or finance departments who would expect to gain revenue from implementing higher minimum prices on alcohol? Depending on how broader or narrow the criteria are will likely impact the number and type of studies included in the final analysis. I would prefer that those studies that meet the other criteria be included,	We excluded any studies which reported a conflict of interest in a 'declaration of interests' or 'acknowledgements' section of the paper. We excluded papers with COIs that would be in favour of (e.g. authors linked to the Alcohol Health Alliance) or against MUP (e.g. industry-funded). We have now clarified this in the methods section. Where it was not possible to determine

even if there was a perceived or reported conflict of interest.	from the paper whether there was a conflict of interest we have classed this as 'not stated' in Table 2.
2. Dissecting minimum pricing. There are instances where the regulated minimum prices are not identical per standard drink, for beer, wine, spirits, coolers. Also, there may be minimum prices, but if pricing is not linked to pure ethanol volume then, for example, stronger beer will be cheaper per unit of pure alcohol than weaker beer. Also, in some cases minimum prices are indexed to inflation, in other cases, not. Of the 35 studies included, I would like to know how many took into account these nuances in their assessment, and how these differences were perceived to impact the outcomes. This would involve going back to the 35 studies, and adding a column to table 2, e.g., between the pricing intervention and outcomes studies column, to provide more detail, as appropriate, where the specific research provided it.	We have clarified in table 2 where minimum prices were indexed to or adjusted for inflation in the 'Pricing intervention studied' column. With the removal of the studies which investigated the effect of price changes (i.e. not a minimum price) recommended by reviewer 1, we do not believe that any of the included studies that examined the effect of a minimum price involved different prices for different drinks or were not linked to ethanol content.
3. Other interventions. This relates to the specificity criteria. I am curious how many of the interventions assessed in the 35 papers only involved alcohol pricing, or if a number also had other policy changes underway during the period under study. I think it would be useful to highlight the latter in the Discussion, and offer some hypothesis if the other interventions enhanced or deflated the outcome.	There are unfortunately not enough examples of where MUP has been implemented in conjunction with other measures to say whether this has a lesser or greater effect. We have noted where other interventions were taking place simultaneously to MUP in the 'Pricing intervention studied' column and also highlighted this in the Discussion, as suggested.
4. Page 3, line 9. For the international audience suggest adding a phrase about the current population of England.	This has been done
5. Page 3, line 23 Suggest adding a line specifically stating why the U.K. government withdrew its support for MUP.	This has been done

VERSION 2 – REVIEW

REVIEWER	Tim Stockwell Centre for Addictions Research of BC University of Victoria I have published papers discussed in the review. Otherwise I do not have any competing interests to declare.
REVIEW RETURNED	02-Feb-2017

GENERAL COMMENTS

General comments

I continue to find this an excellent and timely paper about a pressing public health policy which is under active consideration across multiple jurisdictions. As such, a systematic review of the evidence base specifically focusing on minimum alcohol pricing is a significant contribution. I am also pleased with how the authors have responded to my earlier comments and suggestions, especially modifying their interpretation of one of the Bradford Hill criteria so as to not consider "biological plausibility" but rather examine evidence for economic relationships between the policy and demand for alcohol. I have just a few minor queries to suggest. My suggestion that the review is updated to the end of 2016 may or may not be reasonable - it would of course be ideal but it would be a shame to greatly delay this paper further. I suspect there will be very few new papers published that meet the criteria and it might not take long to do or change the paper very much - other than making it more up-to-date.

Specific comments

1. The literature search was completed for the end of February 2016 which is almost one year ago. If at all possible I would recommend updating to December 2016.

2. I am still a little confused about the criteria for inclusion - the paper is billed as systematic review on minimum pricing but some studies on pricing in general are included and others excluded. Given there is discussion of the relative merits of minimum pricing versus other strategies that increase prices across the board I still think this needs to be thought through more clearly. I do see that the status and relevance of the general pricing studies has been reduced in this draft. I guess I still wonder why they are there at all! Maybe they could be some more justification given the primary focus on minimum pricing?

3. I think it would be good to update the current status of legal challenges in Scotland and specifically mention that the European Court of Justice has been involved because the implications are now span all EU countries. Not a place to discuss the complexities of the full decision but perhaps a sentence or two to note the European context and that the Scottish Court of Sessions decision has been appealed once more by the industry.

4. I note a heavy focus on the introduction on the UK - which I guess may be appropriate for the BMJ but this journal does of course have a very wide international audience. The Republic of Ireland has signalled its intention to introduce MUP, Australia and New Zealand have reviewed doing so as well. Maybe there could be a sentence or two about other countries? There could be brief mention of the very wide range of different types of minimum price in Canada for example.

5. I suggest there should be greater acknowledgement of contextual factors that will determine the strength of the relationship between MUP, consumption and related harms. Perhaps this could go in the limitations. I would want to note the importance of affordability given what we know about differential impacts of minimum prices by income level. Then it is also vital to note that MUP is not a fixed policy for which you can expect fixed outcomes - it can be applied weakly or strongly, broadly or narrowly, in markets which have by other means (e.g. government regulation) determined or not the range of prices and the alcohol market and where prices are already high or not.

6. On P.19 it is noted under "coherence" that the modelling studies and others are mutually supportive inconsistent. I would want to note

	that some of this is inevitable - the modelling studies draw upon the wider literature and apply findings from elsewhere in the models e.g. estimates of the extent to which heavier drinkers cheaper alcohol. 7. I think you overstate the potential disadvantage of taxation for alcohol policy on the basis that increased taxes need not in theory be passed on to the consumer. There has been a comprehensive review discussed by Wagenaar et al, 2009 cited in your paper which shows that the prices are almost invariably passed on to consumers. The real difference is on the breadth of focus and the range of drinks and drinkers affected. Tax increases affect everyone, minimum prices only mostly poorer people who seek out cheap alcohol. 8. P20 line 32-32 – there is a double negative about there not being many studies that don't have good designs. Why not put more positively and state that studies had moderate or strong designs? 9. P21 lines 44-45; what unanswered questions? Be more specific. 10. P21 lines 46-48: need to update appeal by Scotch Whisky association.
--	--

VERSION 2 – AUTHOR RESPONSE

Reviewer 1: Tim Stockwell

General comments		Authors' response
	I continue to find this an excellent and timely paper about a pressing public health policy which is under active consideration across multiple jurisdictions. As such, a systematic review of the evidence base specifically focusing on minimum alcohol pricing is a significant contribution. I am also pleased with how the authors have responded to my earlier comments and suggestions, especially modifying their interpretation of one of the Bradford Hill criteria so as to not consider "biological plausibility" but rather examine evidence for economic relationships between the policy and demand for alcohol. I have just a few minor queries to suggest. My suggestion that the review is updated to the end of 2016 may or may not be reasonable - it would of course be ideal but it would be a shame to greatly delay this paper further. I suspect there will be very few new papers published that meet the criteria and it might not take long to do or change the paper very much - other than making it more up-to-date.	We are very pleased that the reviewer considers our paper excellent and timely, and are grateful to the reviewer for these detailed and helpful comments, which we have dealt with promptly to avoid delay in publication. We hope that we have dealt with these to their satisfaction and that the manuscript is now ready for publication.
Specific comments		
1.	The literature search was completed for the end of February 2016 which is almost one year ago. If at all possible I would	We have now updated the search to 18th February 2017 and updated the

	recommend updating to December 2016.	manuscript accordingly throughout.
2.	I am still a little confused about the criteria for inclusion - the paper is billed as systematic review on minimum pricing but some studies on pricing in general are included and others excluded. Given there is discussion of the relative merits of minimum pricing versus other strategies that increase prices across the board I still think this needs to be thought through more clearly. I do see that the status and relevance of the general pricing studies has been reduced in this draft. I guess I still wonder why they are there at all! Maybe they could be some more justification given the primary focus on minimum pricing?	We have edited the inclusion/exclusion criteria in the methods section and hope that this is now clearer. In response to the previous round of reviewer comments we removed the population level/time series studies about pricing in general. We kept cross-sectional studies about pricing which pertained to the proportion of people drinking alcohol below/above proposed minimum price cut points, but our presentation of these findings was unclear in the table, and it looked like these studies were about pricing in general. We value the reviewer's expertise, and have now edited the table to make it clear these studies show drinking above/below given minimum pricing cut-points. This applies to the following articles: Black 2011, Callinan 2015, Crawford 2012, Falkner 2015 and Sheron 2014. Following this further assessment we have removed Casswell 2014 as this was about pricing in general (split in to high and low), not in relation to a proposed minimum price.
3.	I think it would be good to update the current status of legal challenges in Scotland and specifically mention that the European Court of Justice has been involved because the implications are now span all EU countries. Not a place to discuss the complexities of the full decision but perhaps a sentence or two to note the European context and that the Scottish Court of Sessions decision has been appealed once more by the industry.	We have now updated this to include the European Court of Justice involvement and to update on most recent developments (in introduction section).
4.	I note a heavy focus on the introduction on the UK - which I guess may be appropriate for the BMJ but this journal does of course have a very wide international audience. The Republic of Ireland has signalled its intention to introduce MUP, Australia and New Zealand have reviewed doing so as well. Maybe there could be a sentence or two about other countries? There could be	We have re-written the introduction and this now includes Rol/Australia/New Zealand, followed by an improved and updated description of the UK/Scotland afterwards, including the EU court ruling.

	brief mention of the very wide range of different types of minimum price in Canada for example.	
5.	I suggest there should be greater acknowledgement of contextual factors that will determine the strength of the relationship between MUP, consumption and related harms. Perhaps this could go in the limitations. I would want to note the importance of affordability given what we know about differential impacts of minimum prices by income level. Then it is also vital to note that MUP is not a fixed policy for which you can expect fixed outcomes - it can be applied weakly or strongly, broadly or narrowly, in markets which have by other means (e.g. government regulation) determined or not the range of prices and the alcohol market and where prices are already high or not.	We have added this to the limitations as suggested.
6.	On P.19 it is noted under "coherence" that the modelling studies and others are mutually supportive inconsistent. I would want to note that some of this is inevitable - the modelling studies draw upon the wider literature and apply findings from elsewhere in the models e.g. estimates of the extent to which heavier drinkers cheaper alcohol.	We have rephrased this sentence and noted this as suggested.
7.	I think you overstate the potential disadvantage of taxation for alcohol policy on the basis that increased taxes need not in theory be passed on to the consumer. There has been a comprehensive review discussed by Wagenaar et al, 2009 cited in your paper which shows that the prices are almost invariably passed on to consumers. The real difference is on the breadth of focus and the range of drinks and drinkers affected. Tax increases affect everyone, minimum prices only mostly poorer people who seek out cheap alcohol.	We agree with the reviewer and the sentence mentioning taxation 'pass through' has been removed from the updated introduction and from the analogy section. In the discussion we have also added in a citation to a new Lancet paper which recommended MUP + increased taxation.
8.	P20 line 32-32 – there is a double negative about there not being many studies that don't have good designs. Why not put more positively and state that studies had moderate or strong designs?	We are not sure the page and line numbers in our version quite correspond. Is this in relation to the following sentence - "Only a small minority of studies offered weak support for price-based alcohol policy interventions" ? We have deleted this sentence and stated

		that the vast majority of studies would support these interventions higher up in the same paragraph. We have also clarified in the study strengths paragraph that most studies included were rated as strong or moderate. We hope this is sufficient.
9.	P21 lines 44-45; what unanswered questions? Be more specific.	We have expanded this sentence to explain that we mean unanswered questions in relation to the two Bradford Hill criteria where support was less strong (strength of the association and experiment).
10.	P21 lines 46-48: need to update appeal by Scotch Whisky association.	This has been updated and moved to the introduction of the manuscript.